# Protective role of protease-activated receptor-2 in anaphylaxis model mice

**Maho Nakazawa**[1], **Ryota Tochinai**[2], **Wataru Fujii**[3], **Mao Komori**[1], **Tomohiro Yonezawa**[1], **Yasuyuki Momoi**[1], **Shingo Maeda**[1] *

1 Department of Veterinary Clinical Pathobiology, Graduate School of Agricultural and Life Sciences, The University of Tokyo, Tokyo, Japan, 2 Department of Veterinary Pathophysiology and Animal Health, Graduate School of Agricultural and Life Sciences, The University of Tokyo, Tokyo, Japan, 3 Laboratory of Biomedical Science, Graduate School of Agricultural and Life Sciences, The University of Tokyo, Tokyo, Japan

* amaeda@g.ecc.u-tokyo.ac.jp

**Data Availability Statement:** All data files are available from the Figshare database (Figshare, https://figshare.com/articles/dataset/_b_Protective_role_of_protease-activated_receptor-2_

## Abstract

Anaphylaxis is a severe life-threatening hypersensitivity reaction induced by mast cell degranulation. Among the various mediators of mast cells, little is known about the role of tryptase. Therefore, we aimed to elucidate the role of protease-activating receptor-2 (PAR-2), a receptor activated by tryptase, in murine anaphylactic models using PAR-2-deficient mice and newly generated tryptase-deficient mice. Anaphylaxis was induced by IgE-dependent and IgE-independent mast cell degranulation in mice. PAR-2 deficiency exacerbated the decrease in body temperature and hypotension during anaphylaxis; however, the number of skin mast cells, degree of mast cell degranulation, and systemic and local vascular hyperpermeability were comparable in PAR-2 knockout and wild-type mice. Nitric oxide, which is produced by endothelial nitric oxide synthase (eNOS), is an indispensable vasodilator in anaphylaxis. In the lungs of anaphylactic mice, PAR-2 deficiency promoted eNOS expression and phosphorylation, suggesting a protective effect of PAR-2 against anaphylaxis by downregulating eNOS activation and expression. Based on the hypothesis that the ligand for PAR-2 in anaphylaxis is mast cell tryptase, tryptase-deficient mice were generated using CRISPR-Cas9. In wild-type mice, the PAR-2 antagonist exacerbated the body temperature drop due to anaphylaxis; however, the effect of the PAR-2 antagonist was abolished in tryptase-deficient mice. These results suggest that tryptase is a possible ligand of PAR-2 in anaphylaxis and that the tryptase/PAR-2 pathway attenuates the anaphylactic response in mice.

## Introduction

Anaphylaxis is a severe, life-threatening systemic hypersensitivity reaction characterized by rapid onset of airway, breathing, or circulatory problems [1]. Food, insect venom, and drugs constitute the most common elicitors of anaphylaxis worldwide [1]. Although the mechanism is not fully understood, mast cells and their mediators play a crucial role in the onset of anaphylaxis. Mast cells can be activated by elicitors in an immunoglobulin E (IgE)-dependent or

in_anaphylaxis_model_mice_b_/24630414?file=
43276023, DOI: 10.6084/m9.figshare.24630414).

**Funding:** This study was supported by "Japan Society for the Promotion of Science" JSPS KAKENHI, a Grant-in-Aid for Science Research (Grant Number:19H00968 to SM), and Anicom Capital Research Grant (EVOLVE to SM). The above funders had no role in study design, data collection and analysis, decision to publish, or preparation of the manuscript.

**Competing interests:** The authors have declared that no competing interests exist.

IgE-independent manner and release mediators such as histamine, platelet-activating factor (PAF), prostaglandin D2 (PGD2), and proteases such as tryptase and chymase. These mediators act orchestrally on endothelial and smooth muscle cells, resulting in severe vascular hyperpermeability and vasodilation [2]. Histamine has been reported to increase vascular permeability mainly by nitric oxide (NO)-dependent vasodilation and subsequent increases in blood flow and endothelial barrier dysfunction [3, 4]. In addition, PAF can contribute to both vascular hyperpermeability and vasodilation by activating endothelial nitric oxide synthase (eNOS) and promoting NO production [5]. However, antihistamines have limited effects in patients with anaphylaxis [6] and PAF receptor antagonism cannot completely inhibit hypotension in a rat anaphylaxis model [7], suggesting that the function of known mediators alone does not fully explain the pathogenesis of anaphylaxis and that other mediators are involved.

Tryptase is the most abundant protein in human mast cells [8]. Since tryptase is released by mast cell degranulation, serum tryptase concentration has been reported to correlate with the severity of anaphylaxis and is a useful biomarker for the diagnosis of anaphylaxis [9–11]. However, little is known about the role of tryptase in anaphylaxis pathogenesis. In this study, we focused on protease-activated receptor-2 (PAR-2), a tryptase-activated receptor. PAR-2 is a member of the protease-activated receptor family and is activated by specific serine proteases such as mast cell tryptase, trypsin, factor Xa (FXa), and FVIIa, which cleave the N-terminus of the receptor [12]. PAR-2 is widely expressed in intestinal and airway epithelial cells, endothelial cells, and smooth muscle cells [13]. Activation of PAR-2 in endothelial cells has been reported to modulate VE-cadherin expression and affect vascular barrier function *in vitro* [14]. In addition, PAR-2 appears to be involved in both endothelium-dependent relaxation and contraction [15]. In this study, we investigated the effects of PAR-2 on anaphylactic symptoms in murine models of anaphylaxis.

## Materials and methods

### Ethics statement

All animal experiments were approved by the Animal Care and Use Ethical Committees of the Graduate School of Agricultural and Life Sciences, University of Tokyo (approval no. P19-135 and P20-122H02).

### Regents

Compound 48/80, Evans blue dye, anti-DNP-specific IgE, and DNP-albmin were purchased from Sigma Aldrich (St. Louis, MO, USA). Selective PAR-2 antagonist (ENMD-1068) was purchased from MedChemExpress (Monmouth Junction, NJ, USA).

### Mice

Wild-type (WT) C57BL6/J and ICR mice were purchased from Sankyo Labo Service Corporation (Tokyo, Japan). PAR-2 KO mice were purchased from Jackson Laboratories (Bar Harbor, ME, USA). The tryptases released by mast cells in mice are mouse mast cell protease-6 (mMCP-6) and mouse mast cell protease-7 (mMCP-7) [8]. In humans, δ-tryptase, the ortholog of mMCP-7, is mostly inactive [16]. Wild-type C57BL/6 mice are genetically deficient in *mMCP-7* [17] thus, mMCP-6 deficiency in C57BL/6 mice also implies systemic tryptase deficiency. *mMCP-6* knockout C57BL/6J mice (mMCP-6 KO) were generated by CRISPR/Cas9-mediated genome editing in zygotes, according to previous reports [18, 19].

## Passive systemic anaphylaxis model

Anti-DNP-specific-IgE (clone: SPE-7, 10 μg/head) was intravenously injected into WT and PAR-2 KO mice, followed 18 h later by intravenous injection with DNP-albumin (300 μg/head). The rectal temperature was measured every 5 min for 60 min using a thermometer (Physitemp, NJ, USA).

## IgE-independent anaphylaxis model

C48/80 (2 mg/kg) in 100 μL saline was administered intravenously to the WT, PAR-2 KO, and mMCP-6 KO mice. In some experiments, ENMD-1068 (10 mg/kg) or saline was injected intraperitoneally 30 min before the C48/80 injection. Rectal temperature was measured every 5 min for 60 min.

## Mast cell staining and immunofluorescence staining for mMCP-6 in skin

The mouse ears were fixed in 10% formamide and embedded in paraffin. Sections (2 μm) were deparaffinized and autoclaved at 121˚C for 10 min in 10 mM sodium citrate buffer (pH 6.0). The sections were blocked with 5% skimmed milk in TBS-T at room temperature for 60 min and then incubated with rabbit anti-mMCP-6 antibody (R&D Systems, MN, USA, 1:500) at 4˚C overnight. After washing with TBS, sections were incubated with secondary antibody (Alexa Fluor 594 goat anti-rat IgG, Abcam, 1:500) and FITC-avidin (BioLegend, CA, USA, 1:200) for 1 h at room temperature and counterstained with DAPI. FITC-avidin was used to label mast cells in the skin. Images were captured using a BZ-X800 fluorescence microscope (KEYENCE, Osaka, Japan). The digitized images were transferred to a computer to measure the size of each region using a software (ImageJ, National Institute of Mental Health, MD, USA), and FITC-positive mast cells were counted. The results are expressed as positive cells per mm$^2$.

## Hematocrit, histamine, and mMCP-6 measurement

Blood samples were obtained 10 min after intravenous administration of C48/80 (2 mg/kg) to measure the hematocrit, plasma histamine, and mMCP-6 levels. To determine hematocrit levels, blood was collected in heparinized microhematocrit tubes (DRUMMOND, PA, USA) and centrifuged at 12,000 rpm for 3 min. The ratio of red blood cell volume to total blood volume was determined using a hematocrit reader. Plasma histamine and mMCP-6 levels were measured using a histamine EIA kit (Bertin Bioreagent, Montigny-le-Bretonneux, France) and Mcpt6 ELISA kit (Thermo Fisher Scientific, MA, USA), respectively.

## Vascular permeability assay

To evaluate systemic vascular permeability, blood was collected from WT and PAR-2 KO mice 10 min after vehicle or C48/80 (2 mg/kg) injection. In other experiments, C48/80 (2 mg/kg) with 0.5% Evans blue dye was intravenously administered to WT or PAR-2 KO mice, while the control mice received Evans blue only. Thirty minutes after injection, the lungs were collected and weighed. The extravasated Evans blue was extracted in formamide at 56˚C for 24 h. The concentration of Evans blue was quantified spectrophotometrically at 620 nm using a plate reader (iMark microplate reader, Bio-Rad, CA, USA). To assess local vascular permeability in the skin, C48/80 (100 ng in 20 μL saline, right ear) or saline (left ear) was intradermally injected into the mouse ear. Evans blue (0.5% in 100 μL saline) was injected intravenously 5 min after intradermal anaphylactic stimulation. After 30 min, the ear was dissected and Evans blue leakage was measured as described above.

## Blood pressure measurement

The mice were anesthetized with 1.5% isoflurane and placed in the supine position on a plate heated to 37°C. The left femoral artery was exposed and a polyethylene catheter filled with heparinized saline was inserted. The catheter was connected to a transducer amplifier (Nihon Kohden, Tokyo, Japan) through a pressure transducer (NEC Sanei, Tokyo, Japan), and the arterial pressure was recorded. Mean blood pressure was continuously sampled at intervals of 1 ms using an analysis system (SBP-2000, Softron, Tokyo, Japan) with an analog-to-digital converter connected in series to a personal computer. During the blood pressure recording period, a single dose of C48/80 (2 mg/kg) was administered through the right femoral vein. Blood pressure was recorded for up to 30 min after administration.

## Expression of eNOS mRNA

The lungs of mice were excised before and 1 h after the injection of C48/80 (2 mg/kg). Total RNA was extracted from tissues using NucleoSpin RNA (Takara Bio, Shiga, Japan) according to the manufacturer's instructions. Reverse transcription was performed using a commercially available kit (ReverTra Ace with gDNA remover; TOYOBO, Osaka, Japan). Real-time PCR of eNOS and β-actin was performed using a commercially available kit (THUNDERBIRD Next SYBR qPCR Mix; TOYOBO). The following primer sets were used: eNOS, forward 5'-CAA CGCTACCACGAGGACATT-3' and reverse 5'-CTCCTGCAAAGAAAAGCTCTGG-3'; β-actin, forward 5'-GGAAATCGTGCGTGACATCA-3' and reverse 5'-GCCACAGGATTCCATACCCA-3'. The PCR reaction consisted of 40 cycles, each for 30 s at 95°C, followed by 5 s at 95°C, and 10 s at 60°C. The expression level of eNOS was normalized to that of β-actin as a reference gene, and the values were calculated relative to the baseline of the WT mice, which was converted to 1.

## Western blot

C48/80 (2 mg/kg) was administered intravenously to WT and PAR-2 KO mice, and the lungs were collected before administration and at 10 and 120 min after administration. The lungs were homogenized in RIPA buffer (50 mmol/L Tris-HCl [pH 7.4], 150 mmol/L NaCl, 5 mmol/L EDTA, 1% Triton X-100, 10 mmol/L NaF) containing protease and phosphatase inhibitor cocktails (Wako, Osaka, Japan). For the detection of eNOS and phosphorylated eNOS, the samples (25 μg per lane) were separated by electrophoresis and transferred to PVDF membranes.

The membranes were blocked with 5% BSA at room temperature for 30 min and incubated with anti-eNOS (clone: D9A5L, Cell Signaling Technology, MA, USA, 1:1,000) and anti-Ser177 p-eNOS (clone: C9C3, Cell Signaling Technology, 1:1,000) antibodies. Samples were also probed with an anti-β-actin antibody (GeneTex, CA, USA, 1:1,000) to normalize protein loading. The membrane was then incubated with secondary antibody (donkey anti-rabbit IgG IRDye 680, LICOR Biosciences, NE, USA, 1:10,000) at room temperature for 1 h. The bands were quantified using the Odyssey Imaging System (LICOR Biosciences).

## Statistical analysis

Data are expressed as mean±SEM, and all experiments were repeated at least three times. Statistical analyses was performed using R software (version 4.2.1). Two-way repeated ANOVA was used to compare the degree of temperature reduction between the two groups. For blood pressure comparisons, a two-way repeated ANOVA was performed at two time points: before C48/80 administration and at the time of blood pressure measurement. Skin mast cell counts,

serum histamine and mMCP-6 concentrations, and lung eNOS and phosphorylated eNOS expression were analyzed using Student's *t* test. Vascular permeability and lung eNOS gene expression in anaphylaxis model mice were analyzed using Tukey's test. Statistical significance was defined as $P < 0.05$.

## Results

### PAR-2 deficiency exacerbates IgE-dependent and IgE-independent anaphylaxis in mice without affecting vascular permeability

To verify the effect of PAR-2 on the decrease in body temperature during IgE-dependent anaphylaxis, passive systemic anaphylaxis was induced in the WT and PAR-2 KO mice. IgE-mediated anaphylaxis was more severe in PAR-2 KO mice than in WT mice, as determined by the decrease in body temperature (Fig 1A). Consistent with this finding, intravenous injection of C48/80 induced a more severe decrease in body temperature in PAR-2 KO mice than in WT mice (Fig 1B). These data indicate that PAR-2 deficiency exacerbates the decrease in body temperature during IgE-dependent and IgE-independent anaphylaxis.

Next, we examined the number of mast cells in WT and PAR-2 KO mice. There was no significant difference in the number of mast cells in the skin between WT and PAR-2 KO mice (S1 Fig). To further investigate whether the level of mast cell degranulation differed between WT and PAR-2 KO mice, plasma histamine and mast cell tryptase (mMCP-6) levels were measured after injection of C48/80. There was no significant difference in plasma histamine and mMCP-6 levels between WT and PAR-2 KO mice (S1 Fig).

During anaphylaxis, mast cell mediators such as histamine and PAF induce vascular hyperpermeability, leading to hypotension and hypothermia. We investigated the extent of systemic and local vascular hyperpermeability induced by C48/80 in WT and PAR-2 KO mice. First, the hematocrit was measured in WT and PAR-2 KO mice treated with vehicle or C48/80. In both WT and PAR-2 KO mice, C48/80 injection significantly increased the hematocrit level compared to vehicle injection. However, there was no significant difference in hematocrit between WT and PAR-2 KO mice after vehicle or C48/80 injection (S2 Fig). In addition, PAR-2 deficiency did not affect Evans blue dye leakage in the lungs after systemic C48/80 injection (S2 Fig). Next, we examined the effect of PAR-2 on local vascular permeability after the local

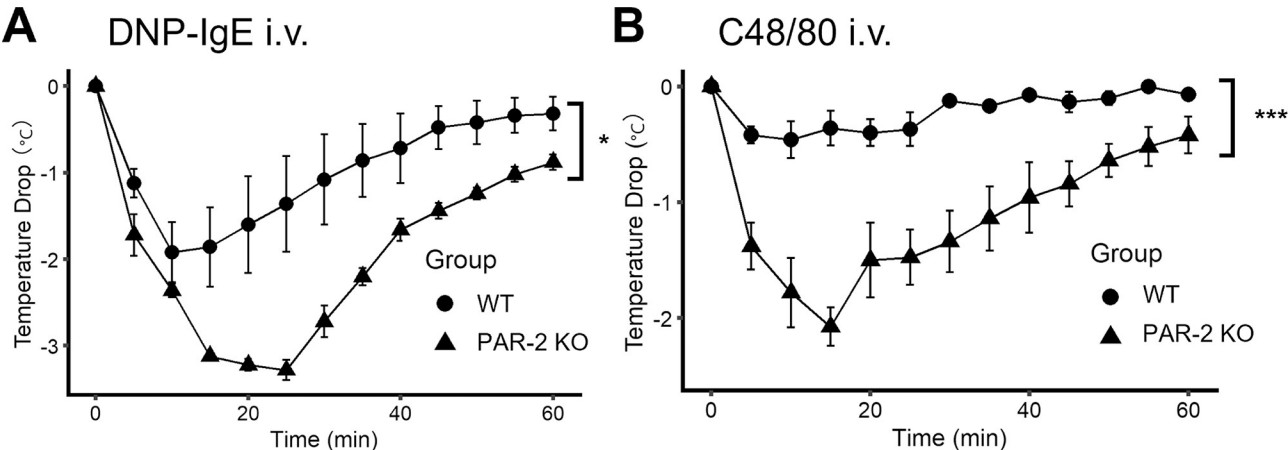

**Fig 1. PAR-2 deficiency exacerbates the body temperature during IgE-dependent and IgE-independent anaphylaxis.** (A) Decrease in body temperature during IgE-dependent anaphylaxis in WT and PAR-2 KO mice (n = 5 each). (B) Decrease in body temperature during IgE-independent anaphylaxis in WT and PAR-2 KO mice (n = 4 each). *$P < 0.05$. ***$P < 0.001$. Data are presented as mean ± SEM.

injection of C48/80 into the auricles of mice. The results showed that Evans blue leakage was significantly increased in C48/80-treated auricles compared to that in solvent-treated auricles, in both WT and PAR-2 KO mice. However, there was no significant difference in the amount of pigment leakage in the auricles of WT and PAR-2 KO mice (S2 Fig). These results suggest that PAR-2 has little effect on systemic and local vascular hyperpermeability induced by C48/80.

## PAR-2 deficiency exacerbates hypotension in mice with anaphylaxis

Hypotension is the hallmark of anaphylaxis. We investigated whether PAR-2 deficiency could affect hypotension induced by C48/80 injection. In WT mice under anesthesia, the mean arterial blood pressure decreased rapidly within 10 min after C48/80 injection and then recovered. PAR-2 KO mice also showed a rapid decrease in blood pressure, but the recovery was slower than that in WT mice and began to recover 30 min after C48/80 injection. Blood pressure in PAR-2 KO mice was significantly lower than that in WT mice 15 and 20 min after C48/80 injection (Fig 2).

## PAR-2 deficiency increases eNOS expression and phosphorylation during anaphylaxis *in vivo*

The decrease in body temperature and hypotension in anaphylaxis are caused by a synergistic effect of increased vascular permeability and decreased vascular resistance due to vasodilation. Previous studies have suggested that PAR-2 deficiency does not affect vascular permeability during anaphylaxis. Therefore, we hypothesized that PAR-2 deficiency may be involved in vasodilation during anaphylaxis. eNOS and its product, NO, have been reported to be important for the decrease in body temperature and hypotension in the anaphylaxis model.

First, gene expression levels in WT and PAR-2 KO mice were evaluated. In WT mice, there was no significant difference in eNOS gene expression in the lungs before and after C48/80 administration. However, in PAR-2 KO mice, eNOS gene expression in the lungs after C48/80 injection was significantly higher than that before C48/80 injection (Fig 3A). Next, eNOS expression and phosphorylation in the lung were quantified by western blotting. At baseline, there was no significant difference in the protein expression and phosphorylation levels of

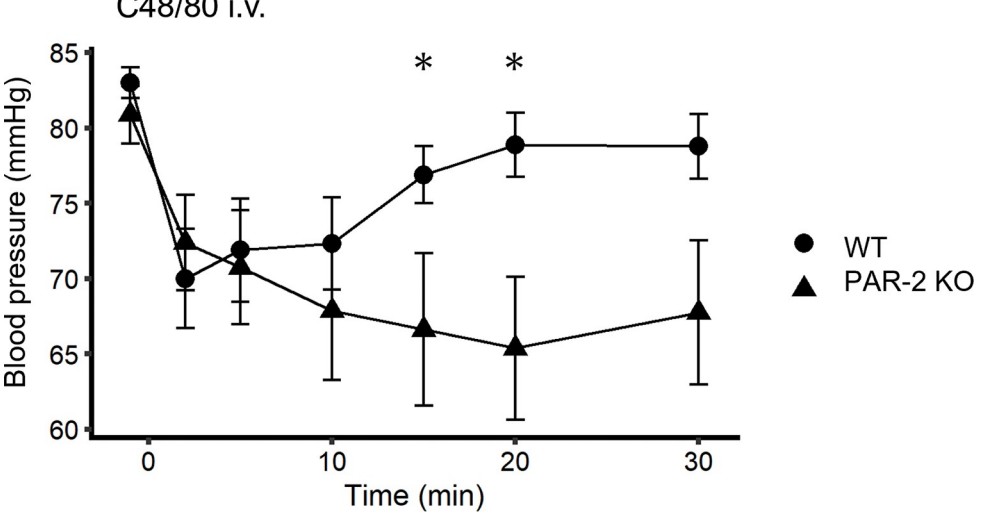

**Fig 2. PAR-2 deficiency exacerbates blood pressure in IgE-independent anaphylaxis.** Arterial blood pressure after injection of C48/80 in WT and PAR-2 KO mice (n = 8–9). *$P < 0.05$. Data are presented as the mean ± SEM.

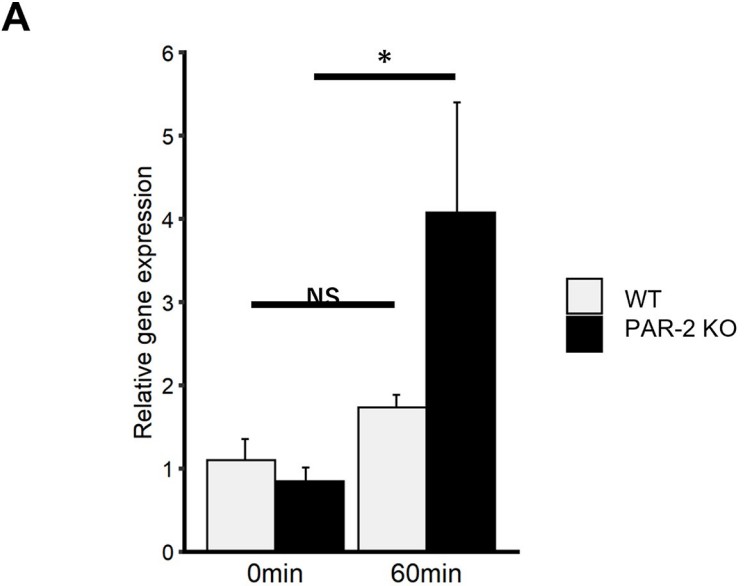

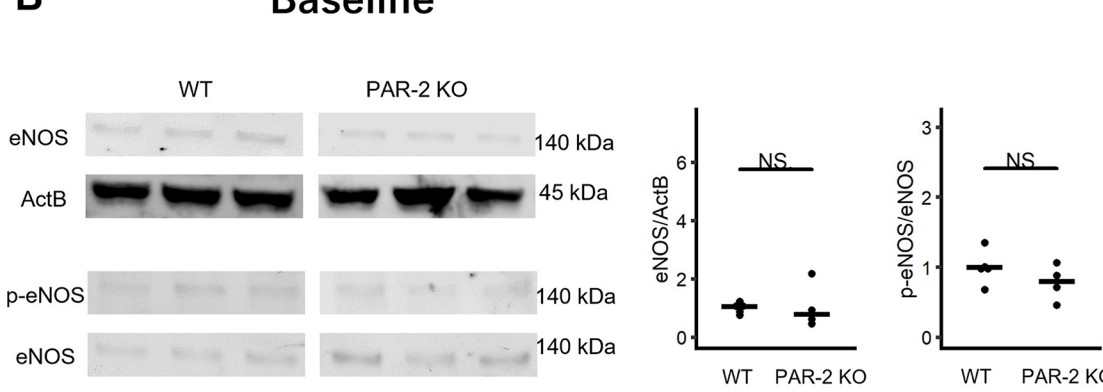

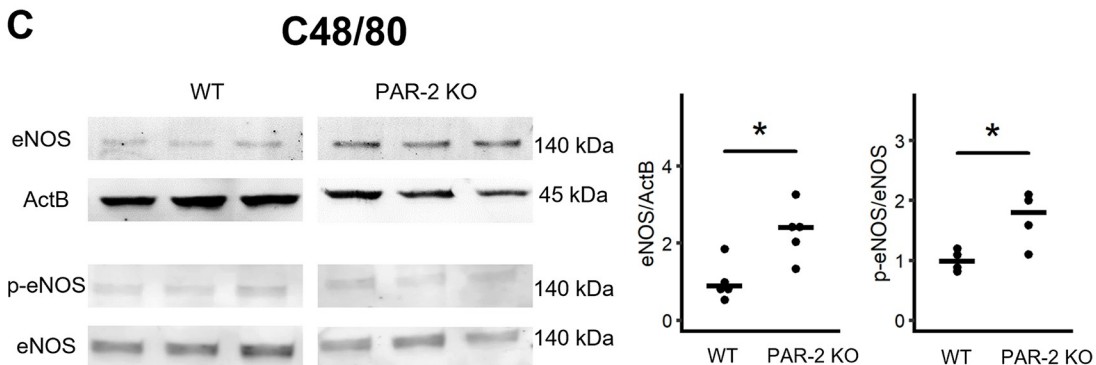

**Fig 3. PAR-2 deficiency promotes eNOS expression and phosphorylation in IgE-independent anaphylaxis.** (A) Relative gene expression of eNOS in the lungs of WT and PAR-2 KO mice before (0 min) and 60 min after (60 min) C48/80 injection (n = 4 each). Data are presented as mean + SEM. (B) Expression and phosphorylation levels of eNOS in lungs of WT and PAR-2 KO mice at baseline (for p-eNOS quantification in PAR-2 KO, n = 4, others, n = 5). eNOS WB of wild type and PAR-2KO mice was performed on separate gels and membranes. In the figure, wild-type and PAR-2KO eNOS bands are illustrated in separate panels.

When comparing the protein expression levels of eNOS between the wild type and PAR-2 KO, the contrast of the membranes for the wild type and PAR-2 KO was the same. (C) Expression and phosphorylation levels of eNOS in lungs of WT and PAR-2 KO mice after injection of C48/80 (for eNOS quantification in WT and PAR-2 KO, n = 5, for p-eNOS quantification, n = 4). NS, not significant. $^{*}P < 0.05$.

eNOS between WT and PAR-2 KO mice (Fig 3B). In contrast, 120 min after C48/80 injection, the expression level of eNOS was significantly higher in PAR-2 KO mice than in WT mice (Fig 3C). Furthermore, 10 min after C48/80 injection, PAR-2 KO mice showed a significantly higher level of eNOS phosphorylation in the lungs than WT mice (Fig 3C). These results suggest that PAR-2 deficiency causes activation of eNOS through phosphorylation and a subsequent increase in eNOS protein expression during anaphylaxis.

## Mast cell protease-6 is a possible ligand for PAR-2 in anaphylaxis

We hypothesized that mast cell tryptase, mMCP-6, is the ligand that activates PAR-2 during anaphylaxis. Therefore, we generated mMCP-6 KO mice using the CRISPR/Cas9 system. Two gRNAs were used to delete exon2 of mMCP-6, which contains the start codon and is essential for mMCP-6 expression (Fig 4A). The designed gRNAs and Cas9 mRNA were then microinjected into C57BL/6J-derived zygotes. The zygotes were then transferred into the oviductal ampulla of 0.5 dpc pseudopregnant ICR females. After obtaining pups by natural birth, the genotypes were confirmed by genome PCR-directed sequencing (Fig 4A). A pup with exon2 deleted and a small indel was backcrossed several times with wild-type C57BL/6J mice, and the offspring were subjected to the following experiments. The phenotype of mMCP-6 KO mice was confirmed by immunofluorescence of the skin. In WT mice, mMCP-6 protein was detected in mast cells (avidin-FITC-positive cells, Fig 4B). In mMCP-6 KO mice, although there was a comparable level of mast cells in the skin as in WT mice, no mMCP-6 expression was observed (Fig 4B). In addition, plasma mMCP-6 concentration was measured in WT mice and mMCP-6 KO mice. mMCP-6 was detected only in WT mice treated with C48/80 whereas no detectable level of mMCP-6 was observed in mMCP-6 KO mice (Fig 4C).

To evaluate the effect of mMCP-6 on anaphylaxis, WT and mMCP-6 KO mice were intravenously injected with C48/80. Contrary to our expectation, there was no significant difference in body temperature decrease between WT and mMCP-6 KO mice (Fig 4C). To test whether mMCP-6 is involved in the suppression of anaphylaxis through PAR-2 activation, we compared the effects of a PAR-2 antagonist (ENMD-1068) on C48/80-induced anaphylaxis in WT and mMCP-6 KO mice. Administration of the PAR-2 antagonist exacerbated the decrease in body temperature during anaphylaxis in WT mice (Fig 5A) whereas administration of the PAR-2 antagonist did not affect the decrease in body temperature (Fig 5C). In addition, WT mice treated with PAR-2 antagonist (ENMD-1068) showed increase of eNOS phosphorylation in lung compared to WT mice treated with vehicle 10 minutes after C48/80 administration (Fig 5B).

## Discussion

The present study on anaphylaxis mouse models provides evidence supporting that PAR-2 has a protective effect on acute decreases in body temperature and hypotension. PAR-2 deficiency did not affect the number of mast cells in the skin or the concentrations of serum histamine and mMCP-6, suggesting that PAR-2 may be involved in the acute response after mast cell degranulation rather than in mast cell degranulation itself. In addition, the results showed that PAR-2 signaling negatively controlled the phosphorylation and production of eNOS during

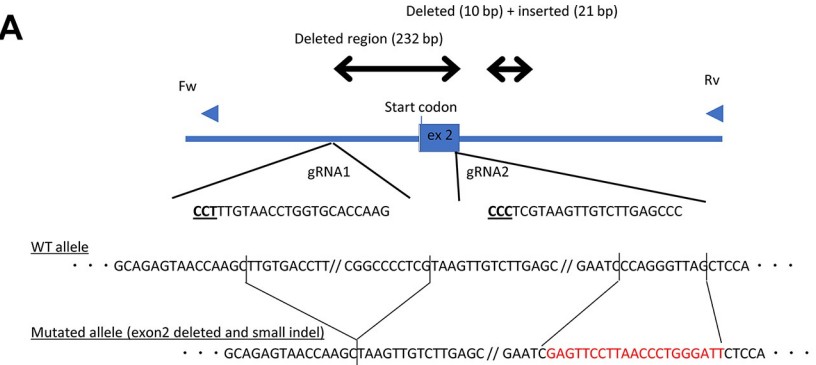

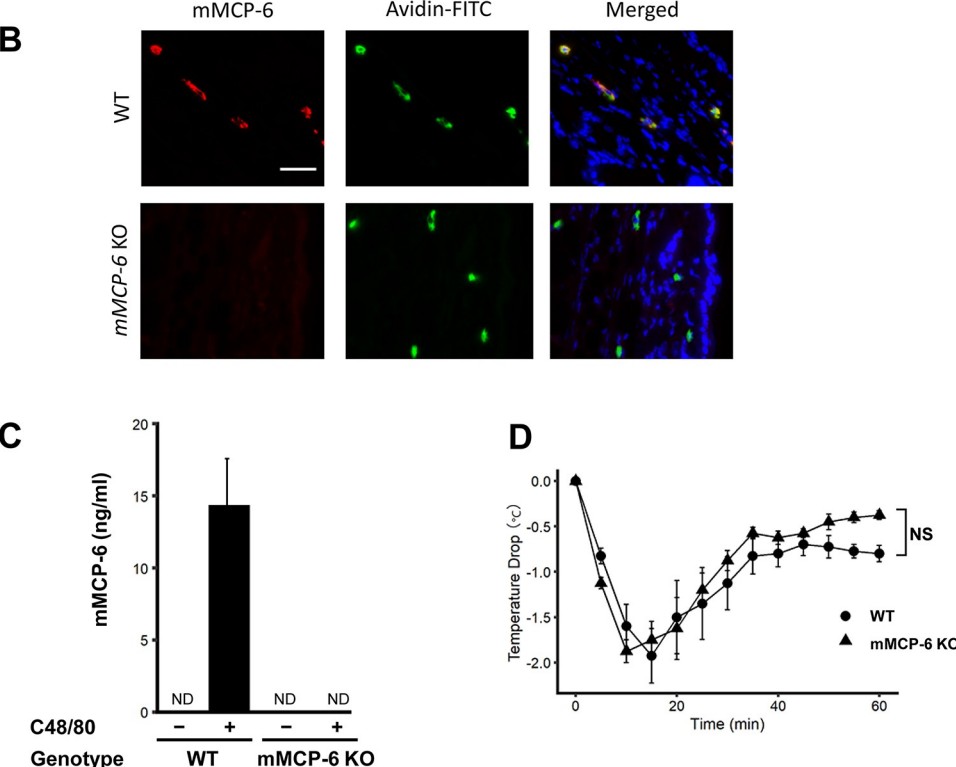

**Fig 4. Creation of mMCP-6 KO mice and the effect of mMCP-6 in anaphylaxis.** (A) Map of mMCP-6 gene and targeting region exon 2. (B) Immunofluorescence of mMCP-6 and mast cell staining in ear skin of WT and mMCP-6 KO mice (magnification, ×400). Bar, 50 μm. (C) Plasma concentration of mMCP-6 in WT and mMCP-6 KO mice 10minutes after vehicle or C48/80 injection (n = 4 each). Data are presented as the mean + SEM. (D) IgE-independent anaphylaxis in WT and mMCP-6 KO mice (n = 5 each). NS, not significant. Data are presented as the mean ± SEM.

anaphylaxis, possibly resulting in the maintenance of body temperature and blood pressure during anaphylaxis. It is highly possible that the ligand of PAR-2 during anaphylaxis is mast cell tryptase, mMCP-6; however, further research is needed to confirm this finding. These results provide novel evidence of the role of PAR-2 in protecting the body from life-threatening anaphylaxis.

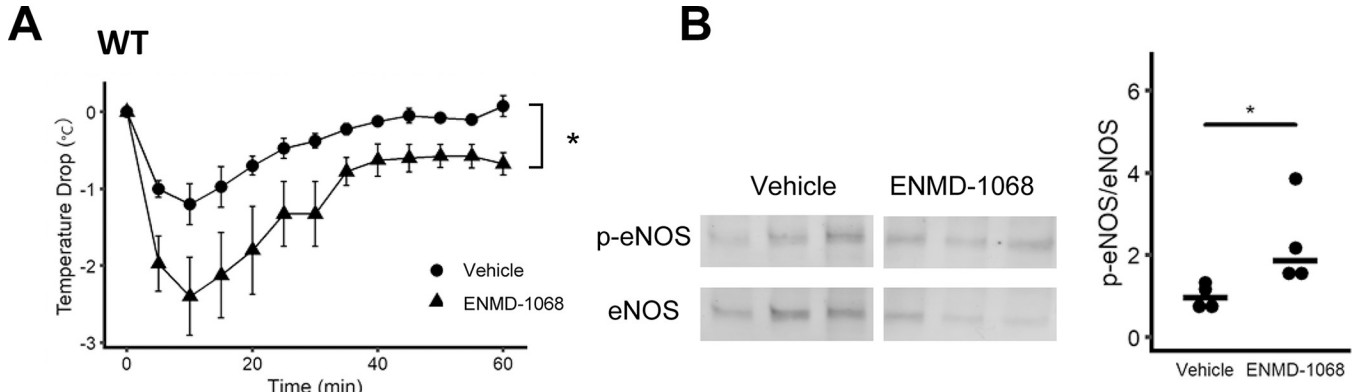

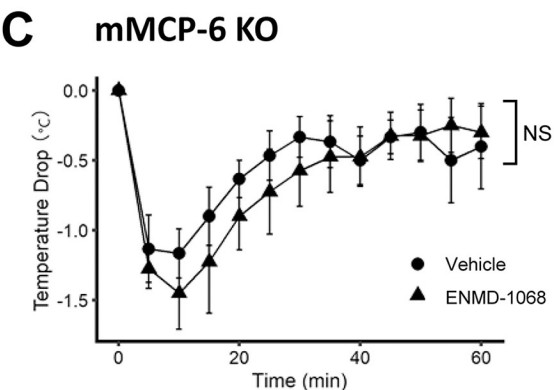

**Fig 5. mMCP-6 deficiency abolishes the effect of the PAR-2 antagonist on anaphylaxis.** (A) The effect of pre-treatment with PAR-2 antagonist in WT mice with IgE-independent anaphylaxis (n = 4 each). Data are presented as the mean ± SEM. (B) Phosphorylation levels of eNOS in lungs of WT-mice pre-treated with vehicle or ENMD-1068 (n = 4 each). (C) The effect of pre-treatment with PAR-2 antagonist in mMCP-6 KO mice with IgE-independent anaphylaxis (n = 4 each).

C48/80 is known to activate mast cells independent of IgE via the Mas-related G protein-coupled receptor b2 (Mrgprb2; mouse ortholog of human MrgprX2) [20]. The two types of mast cell degranulation, IgE-dependent and IgE-independent, differ in their kinetics and duration of mediator release [21, 22]. In the present study, PAR-2 deficiency exacerbated the decrease in body temperature in both IgE-dependent and IgE-independent anaphylaxis, suggesting that PAR-2 is involved in downstream of both IgE-dependent and independent degranulation.

In anaphylaxis, vascular hyperpermeability is an important factor in the exacerbation of symptoms, and several reports using anaphylaxis mouse models have shown a relationship between vascular hyperpermeability and decreased body temperature [23, 24]. Although PAR-2 deficiency exacerbated the symptoms of anaphylaxis, the systemic and local vascular permeability in PAR-2 KO mice was comparable to that in WT mice. These results are consistent with those of asthma and atopic dermatitis models using PAR-2 antagonist and PAR-2 KO mice [25, 26]. Since PAR-2 is thought to have less effect on vascular permeability in anaphylaxis, we focused on vasodilation, which is also crucial for the decrease in body temperature

and hypotension. NO production following eNOS phosphorylation is one of the most important factors for vasodilation [5, 24, 27]. Phosphorylation of eNOS at Ser1177 has been reported to be important for eNOS activation during anaphylaxis in mice [5, 28]. This study showed that PAR-2 deficiency increased the phosphorylation and expression of eNOS *in vivo* after the injection of C48/80. Enhancement of eNOS phosphorylation was reproduced by PAR-2 antagonism. PAR-2 has been reported to be expressed in endothelial cells [13]. Therefore, endothelial PAR-2 may suppress eNOS phosphorylation during anaphylaxis. This study revealed that phosphorylation of eNOS occurred 10 minutes after C48/80 injection in PAR-2 KO mice, followed by mRNA upregulation (60 minutes after injection) and increased eNOS protein expression (120 minutes after injection). Considering that eNOS is produced constitutively in normal condition [29], it is possible that rapid eNOS phosphorylation in early phase and eNOS mRNA and protein expression may be increased compensatory in late phase in PAR-2 deficient mice. Although some reports have shown that PAR-2 activation results in the phosphorylation of eNOS at Ser1177 in human endothelial cells and rat aorta with metabolic syndrome [30–32], another study reported that PAR-2 suppressed eNOS phosphorylation at Ser1177 in the aorta of diabetic mice [33]. Taken together with the results of our study, the effect of PAR-2 on the phosphorylation and activation of eNOS differs depending on the condition of the body. The downstream pathway by which PAR-2 inhibits eNOS phosphorylation remains unclear. It has been reported that phosphatase and tensin homolog deleted on chromosome 10 (PTEN), a $PIP_3$ phosphatase, negatively controls eNOS phosphorylation via Akt, and a rapid decrease in PTEN activity is known to cause an increase in eNOS activation in a murine anaphylaxis model [34]. A previous report showed that PAR-2 activation induces PTEN release and regulates PTEN activity in some cell lines [35]. Further studies are required to clarify the mechanisms by which PAR-2 inhibits eNOS activation during anaphylaxis.

Based on our hypothesis that mast cell tryptase was the ligand for PAR-2 during anaphylaxis, tryptase-deficient (mMCP-6 KO) mice were generated. Contrary to our expectations, mMCP-6 deficiency had no effect on body temperature drop in C48/80-induced anaphylaxis. However, the effect of the PAR-2 antagonist, which exacerbated the decrease in body temperature in the anaphylaxis model, was abolished in mMCP-6 KO mice, suggesting that mMCP-6 may be involved in the inhibitory effect of PAR-2 on anaphylactic symptoms. In addition, these results indicate that the role of mMCP-6 in anaphylaxis is not restricted to that of PAR-2. In humans, tryptase has been reported to have the potential to exacerbate the anaphylaxis by promoting the bradykinin production [36–38].

In conclusion, we showed for the first time that PAR-2 has inhibitory effects on the decrease in body temperature and blood pressure during anaphylaxis, possibly via the upregulation of eNOS phosphorylation and expression. Therefore, PAR-2 agonism may be a potential therapeutic target for anaphylaxis.

## Supporting information

**S1 Fig. Mast cell number and degree of mast cell degranulation in WT and PAR-2 KO mice.** (A) Mast cells stained with FITC-avidin in the ears of mice (left) and the number of mast cells (right, n = 3). Magnification, ×200. Bar, 100 μm. (B) Serum histamine concentrations after C48/80 injection (n = 4). (C) Serum mMCP-6 concentrations after C48/80 injection (n = 4). NS, not significant. Data are presented as the mean ± SEM.
(TIF)

**S2 Fig. The effect of PAR-2 on vascular hyperpermeability induced by mast cell degranulation.** (A) Hematocrit after injection of C48/80 or vehicle in WT or PAR-2 KO mice (n = 4). (B) Evans blue dye leakage in the lungs after systemic injection of C48/80 or vehicle in WT or

PAR-2 KO mice (n = 5). (C) Evans blue dye leakage in the ears after local injection of C48/80 or vehicle in WT or PAR-2 KO mice (n = 5). NS, not significant. ***$P < 0.001$. Data are presented as the mean ± SEM.
(TIF)

**S1 Data. Cropped and uncropped data on western blots in Figs 3B and 3C, and 5B.**
(PDF)

## Acknowledgments

We would like to thank Editage (www.editage.com) for English language editing.

## Author Contributions

**Conceptualization:** Shingo Maeda.

**Data curation:** Maho Nakazawa, Shingo Maeda.

**Formal analysis:** Maho Nakazawa.

**Funding acquisition:** Shingo Maeda.

**Investigation:** Maho Nakazawa, Ryota Tochinai, Wataru Fujii, Mao Komori.

**Methodology:** Ryota Tochinai, Wataru Fujii.

**Project administration:** Shingo Maeda.

**Resources:** Ryota Tochinai, Wataru Fujii.

**Supervision:** Tomohiro Yonezawa, Yasuyuki Momoi, Shingo Maeda.

**Writing – original draft:** Maho Nakazawa, Shingo Maeda.

**Writing – review & editing:** Ryota Tochinai, Wataru Fujii, Mao Komori, Tomohiro Yonezawa, Yasuyuki Momoi, Shingo Maeda.

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
