## [Decision Letter · Decision Letter 0]

6 Jun 2023

PONE-D-23-08225Protective role of protease-activated receptor-2 in anaphylaxis model micePLOS ONE

Dear Dr. Maeda,

Thank you for submitting your manuscript to PLOS ONE. After careful consideration, we feel that it has merit but does not fully meet PLOS ONE’s publication criteria as it currently stands. Therefore, we invite you to submit a revised version of the manuscript that addresses the points raised during the review process.

We look forward to receiving your revised manuscript.

Kind regards,

Hiroyasu Nakano, M.D., Ph.D.

Academic Editor

PLOS ONE

Journal Requirements:

2. Please expand the acronym “JSPS” (as indicated in your financial disclosure) so that it states the name of your funders in full.

"This study was supported by JSPS KAKENHI, a Grant-in-Aid for Science Research (Grant Number:19H00968), and Anicom Capital Research Grant (EVOLVE).."

"This study was supported by JSPS KAKENHI, a Grant-in-Aid for Science Research (Grant Number:19H00968 to SM), and Anicom Capital Research Grant (EVOLVE to SM). The above funders had no role in study design, data collection and analysis, decision to publish, or preparation of the manuscript."

**Additional Editor Comments:**

Although the reviewer feels that the study is potentially interesting, the reviewer has concern about the interpretation of the results. Specifically, an increase in the expression of eNOS at protein levels did not occur prior to the peak temperature drop caused by anaphylaxis. This suggests that eNOS may not be responsible for the temperature drop. Therefore, it is recommended that the authors thoroughly discuss this point in detail.

Reviewers' comments:

Reviewer's Responses to Questions

**Comments to the Author**

1. Is the manuscript technically sound, and do the data support the conclusions?

Reviewer #1: Partly

2. Has the statistical analysis been performed appropriately and rigorously? 

Reviewer #1: Yes

3. Have the authors made all data underlying the findings in their manuscript fully available?

Reviewer #1: Yes

4. Is the manuscript presented in an intelligible fashion and written in standard English?

Reviewer #1: Yes

5. Review Comments to the Author

Reviewer #1: In this article, the authors show that protease-activated receptor-2 (PAR-2) plays a protective role against IgE-dependent and IgE-independent anaphylaxis using PAR-2-deficient mice. In addition, the authors propose that a mast cell tryptase is the ligand for PAR-2 in anaphylaxis and that the tryptase/PAR-2 pathway protects against anaphylaxis by downregulating endothelial nitric oxide synthase (eNOS) activation and expression.

A series of experiments has been conducted in a proper manner. The data presented clearly show that anaphylactic reactions are exacerbated in PAR-2-deficient mice than in wild-type mice. However, I have several concerns about their proposed mechanism by which PAR-2 acts protectively against anaphylaxis.

Major comments,

1. In Figure 3: The authors show that eNOS mRNA expression level was increased at 60 min (Fig. 3A) and eNOS protein expression level was increased at 120 min (Fig. 3C) in the lung of PAR-2-deficient mice after C48/80 administration, respectively. However, the peak temperature drop due to anaphylaxis was 15 min after C48/80 administration (Fig. 1B). If up-regulation of eNOS expression is responsible for the exacerbation of anaphylactic reactions in PAR-2-deficient mice, it likely occurred already before anaphylaxis was induced. Therefore, the authors have to include data or explanations in the manuscript that better illustrate this point.

2. In Figure 4: Increased eNOS phosphorylation levels are a reasonable explanation for exacerbated anaphylactic reactions in PAR-2-deficient mice. Is enhanced eNOS activation also observed when the PAR-2 antagonist ENMD-1068 is administered to wild-type mice? In contrast, is it not observed when that is administered to mMCP-6-deficient mice?

3. In Figure 4: If possible, please also show the time-dependent changes in body temperature during IgE-dependent anaphylaxis in mMCP-6-deficient mice. I think many readers will be interested in that data.

Minor comments,

1. In Figure 4: It would be better to show serum mMCP-6 levels in wild-type mice and mMCP-6-deficient mice during anaphylaxis. In addition to the immunofluorescence images, please provide data that allow quantitative assessment of mMCP-6 gene deficiency.

6. PLOS authors have the option to publish the peer review history of their article (what does this mean?). If published, this will include your full peer review and any attached files.

Reviewer #1: No

---

## [Author Response · Author response to Decision Letter 0]

8 Mar 2024

Dear sir:

Thank you for reviewing our manuscript entitled "Protective role of protease-activated receptor-2 in anaphylaxis model mice": PONE-D-23-08225. We are grateful to you and the reviewers for many helpful suggestions. We also thank you for extending the deadline for the revised manuscript. We carefully read the comments, performed several extra experiments, and revised manuscript. In this document, we described the changes made in response to the reviewers’ comments point-by-point. In order for you to know which parts have been changed, one copy indicates the additions (blue mark). It will be our great pleasure if our manuscript is accepted after the revision. I appreciate again for your kind help.

Sincerely yours,

Shingo Maeda, D.V.M., Ph. D. 

Department of Veterinary Clinical Pathobiology, Graduate School of Agricultural and Life Sciences, The University of Tokyo, 

1-1-1, Yayoi, Bunkyo-ku, Tokyo 113-8657, Japan. 

Tel.: +81-3-5841-3096; fax: +81-3-5841-3096. 

E-mail: amaeda@g.ecc.u-tokyo.ac.jp

 

Response to Reviewers

・1. In Figure 3: The authors show that eNOS mRNA expression level was increased at 60 min (Fig. 3A) and eNOS protein expression level was increased at 120 min (Fig. 3C) in the lung of PAR-2-deficient mice after C48/80 administration, respectively. However, the peak temperature drop due to anaphylaxis was 15 min after C48/80 administration (Fig. 1B). If up-regulation of eNOS expression is responsible for the exacerbation of anaphylactic reactions in PAR-2-deficient mice, it likely occurred already before anaphylaxis was induced. Therefore, the authors have to include data or explanations in the manuscript that better illustrate this point.

Thank you for pointing out the important point. We think that rapid eNOS phosphorylation occurs in early phase of anaphylaxis and decreases body temperature in anaphylaxis. Subsequently, a compensatory increase in eNOS protein expression may occur. We have included this discussion in Manuscript L338 onwards.

・2. In Figure 4: Increased eNOS phosphorylation levels are a reasonable explanation for exacerbated anaphylactic reactions in PAR-2-deficient mice. Is enhanced eNOS activation also observed when the PAR-2 antagonist ENMD-1068 is administered to wild-type mice? In contrast, is it not observed when that is administered to mMCP-6-deficient mice?

Thank you for your very pertinent questions. Additional experiments revealed that PAR-2 antagonism can also enhance eNOS activation in vivo during anaphylaxis (Fig 5B and Manuscript L291 onwards). On the other hand, as there was no difference in body temperature drop during anaphylaxis between mMCP-6 KO mice and WT mice, we believe there must be no difference in eNOS activation between both genotypes. 

3. In Figure 4: If possible, please also show the time-dependent changes in body temperature during IgE-dependent anaphylaxis in mMCP-6-deficient mice. I think many readers will be interested in that data.

We are also interested in that point. There was no different in body temperature drop in IgE-independent anaphylaxis between WT and mMCP-6 KO mice (Fig 4D), suggesting that the role of mMCP-6 in anaphylaxis is not restricted to that of PAR-2. We are also interested in the role of mMCP-6 in anaphylaxis and further experiment, including IgE-dependent anaphylaxis, is necessary.

Minor comments,

1. In Figure 4: It would be better to show serum mMCP-6 levels in wild-type mice and mMCP-6-deficient mice during anaphylaxis. In addition to the immunofluorescence images, please provide data that allow quantitative assessment of mMCP-6 gene deficiency.

Thank you for your accurate comments. We added the result of plasma concentrations of mMCP-6 in WT and mMCP-6 deficient mice (Fig 4C). In WT mice, mMCP-6 was detected after C48/80 injection, on the other hand in mMCP-6 KO mice, mMCP-6 was not detected even in anaphylactic state.

 

Response to Editor’s comments

1. We corrected the raw data of Figure 5B.

As you pointed out, the uncropped data of Fig 5B was wrong one. We replaced this data with correct one. In addition, the annotation of cropped data was also incorrect. The annotation has been corrected. We are very sorry that we made such mistakes.

2. We added the precise number of samples analyzed in Figure 3B and 3C 

In manuscript L260 & 265, we added the precise number of samples analyzed in Figure 3B and 3C.

3.We updated the link to the raw data files in the cover letter.

After updating the raw data, we updated the cover letter link to the raw data. The DOI was also written in cover letter.

---

## [Decision Letter · Decision Letter 1]

27 Mar 2024

Protective role of protease-activated receptor-2 in anaphylaxis model mice

PONE-D-23-08225R1

Dear Dr. Maeda,

We’re pleased to inform you that your manuscript has been judged scientifically suitable for publication and will be formally accepted for publication once it meets all outstanding technical requirements.

Kind regards,

Hiroyasu Nakano, M.D., Ph.D.

Academic Editor

PLOS ONE

Additional Editor Comments (optional):

Reviewers' comments:

Reviewer's Responses to Questions

**Comments to the Author**

1. If the authors have adequately addressed your comments raised in a previous round of review and you feel that this manuscript is now acceptable for publication, you may indicate that here to bypass the “Comments to the Author” section, enter your conflict of interest statement in the “Confidential to Editor” section, and submit your "Accept" recommendation.

Reviewer #1: All comments have been addressed

2. Is the manuscript technically sound, and do the data support the conclusions?

Reviewer #1: Yes

3. Has the statistical analysis been performed appropriately and rigorously? 

Reviewer #1: Yes

4. Have the authors made all data underlying the findings in their manuscript fully available?

Reviewer #1: Yes

5. Is the manuscript presented in an intelligible fashion and written in standard English?

Reviewer #1: Yes

6. Review Comments to the Author

Reviewer #1: (No Response)

7. PLOS authors have the option to publish the peer review history of their article (what does this mean?). If published, this will include your full peer review and any attached files.

Reviewer #1: **Yes: **Nobuhiro Nakano

---

## [Editor Report · Acceptance letter]

8 Apr 2024

PONE-D-23-08225R1 

PLOS ONE

Dear Dr. Maeda, 

I'm pleased to inform you that your manuscript has been deemed suitable for publication in PLOS ONE. Congratulations! Your manuscript is now being handed over to our production team.

Kind regards, 

on behalf of

Professor Hiroyasu Nakano 

Academic Editor

PLOS ONE